# Diagnosing Plantar Plate Injuries: A Narrative Review of Clinical and Imaging Approaches

**DOI:** 10.3390/diagnostics15172188

**Published:** 2025-08-29

**Authors:** Jeong-Jin Park, Hyun-Gyu Seok, Chul Hyun Park

**Affiliations:** 1Department of Orthopedic Surgery, Korea Armed Forces Athletic Corps, Mungyeong 36931, Republic of Korea; wjdwls3912@naver.com; 2Department of Orthopedic Surgery, Armed Forces Capital Hospital, Seongnam 13574, Republic of Korea; rkaldhkthfl4@naver.com; 3Department of Orthopedics, College of Medicine, Yeungnam University, Daegu 42415, Republic of Korea

**Keywords:** plantar plate injury, metatarsophalangeal joint, clinical test, diagnostic imaging, differential diagnosis

## Abstract

Background: Plantar plate injuries represent a common yet frequently underdiagnosed etiology of forefoot pain and metatarsophalangeal joint instability. Diagnostic accuracy is often compromised by nonspecific clinical presentations and significant symptom overlap with other forefoot pathologies, including Morton’s neuroma and synovitis. Early and accurate identification is essential to prevent progression to irreversible deformity. Methods: This narrative review synthesizes recent literature on the clinical evaluation, imaging modalities, and differential diagnosis of plantar plate injuries. A comprehensive literature search in a narrative review format of key databases and relevant journals was performed to critically appraise the diagnostic accuracy, advantages, limitations, and clinical implications of various diagnostic techniques. Results: Physical examination maneuvers—including the drawer test, toe purchase test, and Kelikian push-up test—provide important diagnostic insights but are constrained by operator dependency and lack of standardization. Among imaging modalities, MRI and dynamic ultrasound offer high diagnostic utility, with MRI providing superior specificity and ultrasound enabling functional, real-time assessment. Emerging techniques such as dorsiflexion-stress MRI and dual-energy CT show promising diagnostic potential, though broader clinical validation is lacking. Differential diagnosis remains a major challenge, given the clinical and radiological similarities shared with other forefoot conditions. Conclusions: Accurate diagnosis of plantar plate injuries necessitates a multimodal strategy that combines clinical suspicion, structured physical examination, and advanced imaging. Acknowledging the limitations of each diagnostic modality and integrating findings within the broader clinical context are essential for timely and accurate diagnosis. Future research should prioritize validation of diagnostic criteria, enhanced access to dynamic imaging, and the development of consensus-based grading systems to improve diagnostic precision and patient outcomes.

## 1. Introduction

Plantar plate injury is an increasingly recognized yet frequently underdiagnosed condition characterized by pain and instability of the lesser metatarsophalangeal (MTP) joints, most commonly affecting the second toe [1,2]. The plantar plate is a fibrocartilaginous structure that provides both static and dynamic stability to the MTP joints, counteracting dorsal subluxation and facilitating appropriate load distribution during gait [1,2,3]. Due to its anatomical position and biomechanical function, the plantar plate is particularly vulnerable to repetitive forefoot overload and acute trauma, especially in athletes and individuals with altered foot biomechanics [4,5].

Clinically, plantar plate injuries present with nonspecific findings, including localized plantar forefoot pain, subtle swelling, and progressive deformity [1,4]. These manifestations frequently overlap with other forefoot pathologies such as Morton’s neuroma, capsulitis, Freiberg disease, and stress fractures, complicating accurate diagnosis [6,7]. Moreover, physical examination alone demonstrates significant variability in sensitivity and specificity, underscoring the need for a more systematic diagnostic framework that incorporates advanced imaging techniques [8,9].

Magnetic resonance imaging (MRI) and dynamic ultrasound (US) have greatly advanced the detection of plantar plate pathology [8,10,11]. MRI remains the reference standard owing to its superior soft tissue contrast and multiplanar imaging capabilities [8,11,12,13]. However, recent studies also highlight the diagnostic utility of dynamic US, which offers functional assessment and greater accessibility, making it particularly useful for initial evaluation and confirmation [14,15]. Both modalities, nonetheless, have limitations: MRI may underestimate subtle or partial tears, and the diagnostic accuracy of US is highly dependent on operator expertise and technique [15,16].

This narrative review aims to consolidate current evidence on the diagnostic evaluation of plantar plate injuries by critically appraising the strengths, limitations, and complementary roles of clinical and imaging-based assessment methods. Additionally, it seeks to clarify key diagnostic challenges and provide practical insights to support more accurate and timely clinical decision-making.

To achieve these aims, the review was informed by a literature search of PubMed, Embase, and Scopus databases from their inception through April 2025. The search combined the following terms and their variations: “plantar plate,” “metatarsophalangeal joint instability,” “drawer test,” “toe purchase test,” “Kelikian push-up test,” “crossover toe,” “US,” and “MRI” Eligible studies were English-language publications addressing the diagnosis of plantar plate injuries using clinical examination or imaging methods. Case reports, conference abstracts, and purely surgical technique articles without diagnostic assessment were excluded. Additional references were identified by manually reviewing the bibliographies of included articles and relevant review papers. Although this review did not follow a systematic protocol, efforts were made to incorporate a broad range of high-quality studies to provide a balanced and comprehensive overview of current diagnostic approaches.

## 2. Anatomy and Biomechanics of the Plantar Plate

The plantar plate is a fibrocartilaginous structure located on the plantar aspect of the MTP joints. Composed predominantly of type I collagen fibers, it confers both structural integrity and functional flexibility to the joint complex [17,18]. Anatomically, the plantar plate originates proximally from the periosteum of the metatarsal neck and inserts distally onto the plantar base of the proximal phalanx [4,17]. It also merges medially and laterally with the collateral ligaments, forming an essential stabilizing complex for the MTP joints [19,20].

Functionally, the plantar plate acts as a primary stabilizer by preventing dorsal displacement of the proximal phalanx during the stance phase of gait, thereby maintaining proper alignment and load distribution across the forefoot [5,17]. It counterbalances the dorsiflexion moment generated by ground reaction forces and intrinsic muscle contractions, particularly at the second MTP joint, which endures the greatest peak pressure due to its anatomical length and positioning [21,22].

This structure is especially prone to injury under conditions of repetitive mechanical stress or chronic overload [2,4]. Acute injuries may occur following sudden hyperextension of the MTP joint during sports or accidental trauma, while chronic overloading can lead to gradual attenuation of the plantar plate [2,4]. Contributing factors include biomechanical abnormalities such as metatarsal length discrepancies, excessive pronation, and the use of high-heeled footwear, all of which elevate plantar forefoot pressure and predispose the plantar plate to progressive degeneration and rupture [15,23,24]. In addition, systemic conditions such as rheumatoid arthritis, diabetes mellitus, and generalized ligamentous laxity may further compromise connective tissue integrity and joint stability, increasing susceptibility to injury [17,25]. Understanding the anatomical configuration and biomechanical function of the plantar plate is therefore critical for accurate diagnosis and the formulation of effective treatment strategies [26,27].

## 3. Clinical Examination

Evaluation of plantar plate injury should begin with a thorough patient history and careful visual inspection of the forefoot [28]. Patients typically report pain localized to the plantar aspect of the second MTP joint, often exacerbated by barefoot ambulation or toe dorsiflexion [17,29]. On inspection, subtle signs such as swelling, plantar ecchymosis, or an emerging hammertoe deformity may be observed [17]. As the condition progresses, more overt deformities—including MTP joint subluxation or dislocation—may become apparent [26].

Several clinical maneuvers have been described to facilitate the diagnosis of plantar plate tears.

### 3.1. Drawer Test (Vertical Lachman Test)

The drawer test, also known as the vertical Lachman test, is a key physical examination maneuver used to assess sagittal plane instability of the lesser MTP joints, specifically targeting the structural integrity of the plantar plate [9,17]. The technique involves stabilizing the metatarsal head with one hand while applying a dorsal translational force to the base of the proximal phalanx with the other [9]. A positive test is indicated by excessive dorsal displacement or a lack of a firm endpoint, suggestive of plantar plate insufficiency [9,17,25].

In a surgical cohort, Klein et al. reported a sensitivity of 80.6% and an exceptional specificity of 99.8% for the drawer test in identifying plantar plate tears, underscoring its diagnostic utility [9]. They noted that a positive test was strongly predictive of structural damage, whereas a negative result correlated with the absence of significant tears [9].

Nery et al. further validated the test’s clinical value through correlation with arthroscopically confirmed injuries [30]. Their prospective study identified the drawer test as the most reliable and accurate physical examination parameter for predicting the anatomical severity of plantar plate tears [30].

Supporting these findings, cadaveric data from Bergeron et al. demonstrated that the sensitivity of the drawer test increases with lesion size [31]: detectable instability was present in 66.7% of specimens with 2 mm tears, 86.7% with 4 mm tears, and 100% with tears measuring ≥6 mm [31].

Despite its strengths, the drawer test is not without limitations. Bergeron et al. also reported a false-positive rate of 16.7%, potentially attributable to anatomical variation or underlying joint laxity [31]. Furthermore, Fernandes et al. found that 34.4% of asymptomatic individuals with normal US findings exhibited a mildly positive drawer test, highlighting the possibility of subclinical hypermobility or misinterpretation [32].

Operator dependency is another critical limitation. Variations in applied force, hand positioning, and subjective assessment of endpoint resistance can result in inconsistent findings, particularly among less experienced clinicians [9,30]. Additionally, early-stage or partial tears may not produce appreciable mechanical instability, thereby reducing the test’s sensitivity in these cases [31].

In summary, the drawer test is a valuable and specific clinical tool for detecting plantar plate instability, especially in the context of high-grade or complete tears. It is recommended as a fundamental component of the clinical evaluation for patients with suspected plantar plate pathology. However, due to its susceptibility to false positives, interobserver variability, and reduced performance in early disease, the test should be interpreted in conjunction with imaging studies and additional clinical findings to ensure diagnostic accuracy.

### 3.2. Toe Purchase Test (Paper Pullout Test)

The toe purchase test, also referred to as the paper pullout test, is used to assess dynamic plantarflexion strength and digital ground contact [25,26]. In this test, a narrow strip of paper is placed beneath the affected toe while the patient is instructed to press the toe actively against the ground [25]. The examiner then attempts to withdraw the paper. Inability to resist the pull is considered a positive result, indicating reduced digital purchase and suggesting plantar plate insufficiency [25,26]

This test is particularly useful for identifying early to moderate stages of plantar plate dysfunction, in which joint alignment may still appear intact, yet effective toe grip is impaired [26]. Multiple studies have reported correlations between positive toe purchase test findings and intraoperative confirmation of plantar plate tears [25,26,27,29]. Despite its simplicity and clinical applicability, the test remains qualitative in nature, with reproducibility limited by examiner-dependent variables such as pulling force and patient-specific differences in toe strength [25].

### 3.3. Kelikian Push-Up Test

The Kelikian push-up test involves applying upward pressure to the plantar surface of the metatarsal head while observing for elevation or rigidity of the corresponding toe [33,34,35]. This maneuver is primarily used to assess joint reducibility and evaluate the integrity of soft tissue structures, including the plantar plate and collateral ligaments [33,34].

Although frequently employed in the clinical evaluation of crossover toe and MTP joint instability, the test remains largely subjective. It lacks standardized diagnostic thresholds and has not been formally validated in terms of accuracy [33,34]. Therefore, its clinical value remains primarily qualitative and experiential rather than evidence-based. Nevertheless, it is widely regarded as a helpful adjunct in surgical settings, particularly for assessing the flexibility of toe deformities and aiding in operative planning [26,34].

### 3.4. Crossover Toe Deformity

Crossover toe deformity, while not a provocative test per se, is routinely evaluated during clinical examination due to its diagnostic significance [9]. It reflects a chronic manifestation of MTP joint instability and serves as a key visual marker of advanced plantar plate pathology [9,36].

This deformity is characterized by dorsomedial deviation and displacement of the second toe, typically resulting from chronic attenuation or rupture of the plantar plate and medial collateral ligament complex [9,27]. It is generally considered a late-stage outcome of longstanding joint instability [9,27].

Clinically, crossover toe deformity is highly specific (88.9%) for significant plantar plate injury but demonstrates low sensitivity (8.0%), making it a strong indicator of advanced disease but a poor screening tool for early pathology [9]. It is often accompanied by additional signs such as loss of toe purchase, a positive drawer test, and visible joint subluxation or dislocation [9]. Akoh et al. noted that this deformity is most commonly associated with high-grade plantar plate injuries (grade III or IV), reflecting a failure of both static and dynamic stabilizing structures [27].

### 3.5. Summary

Physical examination remains a cornerstone in the evaluation of plantar plate pathology. The drawer test demonstrates high specificity and sensitivity, particularly for identifying advanced tears or gross joint instability. The toe purchase test serves as a valuable adjunct for assessing dynamic plantarflexion strength and is capable of detecting subtle functional deficits in early disease stages. Although the Kelikian push-up test lacks standardization, it offers practical utility in evaluating deformity reducibility and informing surgical decisions. Crossover toe deformity is a hallmark of late-stage plantar plate injury and strongly suggests high-grade ligamentous compromise. Given the variability in test performance and interpretation, clinical examination findings should be integrated with imaging results to ensure a comprehensive and accurate diagnosis.

The diagnostic performance, clinical relevance, and limitations of each clinical test described above are summarized in Table 1 to provide a concise reference for their comparative utility.

## 4. Imaging Modalities

Accurate imaging plays a central role in the diagnosis of plantar plate injuries, particularly in cases where clinical findings are inconclusive [10,23]. A variety of imaging modalities—including plain radiographs, US, and MRI—provide complementary information regarding the morphology, structural integrity, and functional dynamics of the plantar plate complex [10].

### 4.1. Plain Radiographs

Weight-bearing plain radiographs are frequently employed as an initial diagnostic tool in patients presenting with forefoot pain and suspected plantar plate pathology [37]. Although direct visualization of the plantar plate is not possible with this modality, radiographs may reveal indirect signs such as MTP joint subluxation, angular deformities, and altered metatarsal parabola alignment [37].

Mann et al. identified an association between increased second metatarsal protrusion and plantar plate rupture, suggesting that osseous length discrepancies may predispose the joint to biomechanical overload and instability [22]. Likewise, Klein et al. reported that radiographic findings such as second toe splaying and exaggerated metatarsal protrusion were more commonly observed in patients with confirmed plantar plate injuries [3].

Despite these observations, the diagnostic utility of plain radiographs is limited, especially in early or partial plantar plate tears, due to their inability to directly assess soft tissue structures [3,22,38]. Therefore, radiographic findings should be interpreted with caution and in conjunction with clinical assessment and advanced imaging techniques [22,38].

### 4.2. Ultrasound (US)

Musculoskeletal US is widely utilized as a first-line imaging modality for suspected plantar plate injuries, owing to its non-invasiveness, widespread availability, and capacity for dynamic assessment [8,39]. High-frequency linear transducers allow visualization of the plantar plate as a hyperechoic band in the longitudinal plane; pathological findings may include hypoechoic clefts, focal thinning, or discontinuity [40,41].

Dynamic US—performed during passive dorsiflexion or simulated loading of the toe—enhances diagnostic sensitivity, particularly for identifying partial or functional tears that may not be apparent on static imaging [42]. In cadaveric validation studies, Stone et al. demonstrated that US findings corresponded closely with both MRI and histological evaluations, confirming its reliability under controlled conditions [43].

Clinical studies further support the utility of US. Gregg et al. reported a sensitivity of 96% and a specificity of 87% for US in detecting plantar plate tears, using surgical findings as the reference standard [44]. Klein et al. also found that US accurately identified 41 of 45 confirmed tears in a preoperative cohort, although four full-thickness lesions were missed—likely due to technical limitations or tear orientation [21].

Carlson et al. confirmed the feasibility of using US in routine clinical settings, but emphasized that diagnostic accuracy is closely tied to examiner experience [40]. Similarly, McCarthy et al. highlighted the potential for false positives and false negatives in less experienced hands or under suboptimal technical conditions [39].

Despite its advantages—including affordability, real-time functional imaging, and accessibility—US remains operator-dependent and may be affected by factors such as patient body habitus, joint depth, and anisotropy artifacts [39,41].

In summary, US represents a valuable and practical modality for the evaluation of plantar plate pathology, particularly when dynamic techniques are employed and interpretation is performed by experienced clinicians. Nonetheless, in cases of diagnostic uncertainty or when comprehensive preoperative evaluation is required, MRI remains indispensable for defining the extent and precise location of injury [8,42].

### 4.3. Magnetic Resonance Imaging

MRI is widely considered the reference standard for evaluating plantar plate pathology due to its superior soft tissue contrast, multiplanar imaging capabilities, and ability to delineate both direct and indirect signs of injury [12,41]. It demonstrates high diagnostic accuracy, particularly when interpreted alongside clinical findings [15,45], and plays a critical role in preoperative planning by providing detailed structural assessment [45].

On MRI, normal plantar plates appear as uniformly low-signal structures across all sequences, exhibiting smooth margins on sagittal and coronal proton density and T2-weighted fat-suppressed images [12,41]. Direct signs of plantar plate tear include hyperintensity and discontinuity at the proximal phalangeal insertion, typically best visualized in the sagittal plane [41]. Additional features such as plate retraction and adjacent bone marrow edema may also be observed [19,41]. Indirect signs—such as pericapsular fibrosis and capsular thickening—can resemble Morton’s neuroma and must be carefully differentiated [15,19].

Nery et al. emphasized the utility of anatomical grading systems to improve diagnostic performance, reporting an increase in radiologist accuracy from 77.0% to 88.5% when the system was applied prior to interpretation [12].

In a comparative study using surgical findings as the reference standard, Yamada et al. reported that MRI achieved a sensitivity of 90.9% and specificity of 79.3% for full-thickness tears when both direct and indirect signs were considered [46]. However, reliance on the direct sign of discontinuity with fluid interposition alone yielded a sensitivity of only 38.2%, underscoring the importance of comprehensive interpretation strategies [46].

MRI also facilitates the identification of degenerative or partial-thickness tears, which may manifest as subtle signal alterations or thickening on T2-weighted images. In a retrospective analysis, short-axis intermediate-weighted sequences achieved a sensitivity of 92% for detecting degenerative plantar plate changes, though specificity was limited to 61.3% [22]. This limited specificity may reflect signal changes related to age-associated degeneration or adjacent synovitis that do not necessarily correspond to a clinically significant tear [24,28]. Umans et al. noted that early dysfunction may be inferred from pericapsular fibrosis and signal abnormalities, even in the absence of frank discontinuity [13,47].

Recent advances include MRI stress testing, in which the affected toe is held in hyperextension during scanning [45]. Giuliani et al. demonstrated that this technique improved diagnostic sensitivity by identifying tears in two symptomatic patients who had normal MRI findings in neutral positioning. Additionally, this approach revealed subtle dorsal subluxation in cases of occult instability [45].

Despite its diagnostic strengths, MRI has notable limitations. Nery et al. reported only fair interobserver agreement without prior training in the grading system, indicating variability in interpretation among radiologists [12]. Moreover, MRI may exhibit limited sensitivity for partial-thickness or lower-grade tears, especially in early disease stages [48]. Subtle lesions may also be underestimated, with diagnostic performance influenced by factors such as image resolution and the examiner’s level of experience [49].

In summary, MRI provides excellent specificity and detailed structural visualization, making it the most comprehensive imaging modality for diagnosing plantar plate injuries and facilitating surgical planning. Incorporation of anatomical grading systems and stress-positioning techniques may further enhance its diagnostic value. Nevertheless, MRI findings should always be interpreted within the clinical context, and additional evaluation with US or surgical correlation may be necessary for subtle or equivocal cases.

### 4.4. Comparison of US and MRI for Diagnosing Plantar Plate Injury

Several comparative studies have assessed the diagnostic performance of US and MRI in detecting plantar plate pathology [8,15,16,40]. While both modalities are clinically valuable, they differ in sensitivity, specificity, accessibility, and operator dependency [8,15,16,40].

US has demonstrated high diagnostic accuracy, particularly when using high-frequency transducers and dynamic scanning protocols [39,40,41,50]. In cadaveric and small surgical cohort studies, US has achieved near-perfect sensitivity and specificity for plantar plate tears [8,40]. Dynamic US, in particular, enhances the ability to detect subtle joint instability and partial tears that may be missed with static imaging [42]. However, the accuracy of US is highly dependent on operator skill and can be affected by technical factors such as anisotropy artifacts and patient-specific anatomical variables [21].

MRI, in contrast, offers more standardized image acquisition and interpretation with less operator dependence. It provides superior soft tissue contrast and comprehensive structural evaluation [11,45]. While some studies have reported marginally lower sensitivity compared to US, MRI typically offers higher specificity and is considered more reliable across a variety of clinical settings [14,16].

A recent meta-analysis by Albright et al. confirmed the high diagnostic value of both modalities [15]. US provides dynamic assessment at a lower cost and with broader availability, while MRI affords greater consistency and remains the preferred modality for surgical planning or when diagnostic uncertainty persists [15].

### 4.5. Dual-Energy Computed Tomography (DECT)

Dual-energy computed tomography (DECT) is an advanced imaging technique that enables visualization of collagen-rich structures through material decomposition algorithms, producing collagen-sensitive color maps without additional radiation exposure [51,52]. This capability allows tissue characterization beyond conventional grayscale CT, offering advantages such as rapid image acquisition, material-specific mapping, and reduction in metal artifacts [52,53].

In orthopedic practice, DECT has been successfully applied to detect anterior and posterior cruciate ligament tears with high diagnostic accuracy, grade degenerative intervertebral disc changes with excellent inter-reader reliability, and identify urate crystal deposition in gout [52,53,54]. It has also been used to visualize cervical and wrist ligaments, detect bone marrow edema, and improve evaluation in the presence of orthopedic hardware, demonstrating its versatility across musculoskeletal conditions [55,56,57]. In the context of plantar plate injury, evidence is currently limited to isolated case reports [51]. For example, Stevens et al. described a case in which DECT identified a plantar plate tear using collagen-specific color mapping, with findings confirmed by MRI [51]. While these results suggest potential for detailed soft-tissue assessment, its clinical role in plantar plate evaluation remains preliminary, as it has not yet been validated in larger, controlled clinical studies.

### 4.6. Summary

Each imaging modality offers unique advantages and limitations in the evaluation of plantar plate injuries. Plain radiographs may reveal secondary signs of pathology—such as subluxation or angular deformities—but do not permit direct visualization of soft tissue structures. US provides high sensitivity and dynamic functional assessment but is limited by operator dependency and reduced specificity. MRI remains the gold standard due to its high specificity, reproducibility, and ability to delineate both full-thickness and degenerative tears with excellent anatomical detail. DECT is an emerging modality with potential for visualizing collagen loss in plantar plate injuries, though it remains investigational at this stage. Selection of imaging should be guided by the clinical context, with US serving as a useful screening tool and MRI preferred for definitive diagnosis and surgical planning.

The diagnostic performance, clinical relevance, and limitations of each imaging modality described above are summarized in Table 2 to provide a concise reference for their comparative utility.

## 5. Differential Diagnosis of Plantar Plate Injury

Plantar plate injury typically presents with forefoot pain localized beneath the second or third MTP joint and is often accompanied by swelling, joint instability, or toe deviation [1,2,4]. However, several other conditions share overlapping clinical features and must be carefully distinguished through detailed physical examination and appropriate imaging studies [6,7].

Morton’s neuroma is a common entrapment neuropathy involving the common digital nerve, most frequently located between the third and fourth metatarsals [6]. It typically manifests as sharp, burning pain or numbness localized to the intermetatarsal space, which is often aggravated by tight footwear or prolonged standing [6]. Mulder’s sign—a palpable or audible click elicited by compressing the forefoot—may be present during examination [6,7]. On US, Morton’s neuroma appears as a well-circumscribed hypoechoic lesion within the intermetatarsal space [7,58]. In contrast to plantar plate injury, which presents with plantar tenderness over the MTP joint and sagittal plane instability, neuroma-related pain is neuritic and localized to the webspace [6,58]. Notably, coexistence of Morton’s neuroma and plantar plate insufficiency is not uncommon, necessitating a comprehensive and nuanced diagnostic approach [26].

MTP joint synovitis, particularly in patients with rheumatoid arthritis, may mimic plantar plate pathology due to similar findings such as joint swelling, pain, and deformity [7,9,23]. However, synovitis in this context is typically bilateral, involves multiple joints, and is associated with systemic features [27]. On US, synovitis appears as hypoechoic synovial thickening with increased Doppler signal, while MRI typically reveals joint effusion, synovial enhancement, and periarticular erosion [7,18,58]. Clinical differentiation is further aided by findings such as morning stiffness, elevated inflammatory markers, and characteristic radiographic erosions [7,59].

Freiberg disease is a form of osteonecrosis affecting the metatarsal head, most commonly involving the second ray in adolescents and young adults [7,58]. It presents as activity-related dorsal forefoot pain and may be accompanied by reduced range of motion [7]. Radiographic findings include sclerosis, flattening, or fragmentation of the affected metatarsal head, while early disease may be identified on MRI by the presence of subchondral collapse or bone marrow edema [7,18,58]. Unlike plantar plate injury, Freiberg disease does not typically present with plantar pain, joint instability, or toe deviation [7].

Early capsular inflammation or generalized ligamentous laxity may present with vague MTP joint discomfort and mild swelling, without the structural deformities characteristic of plantar plate rupture [60]. These patients typically do not exhibit instability on drawer testing, and imaging studies reveal no discrete plantar plate tear [60]. Doty and Coughlin proposed that such cases may represent early-stage plantar plate insufficiency, which can progress to complete rupture if not identified and addressed in a timely manner [26]. Differentiation is based on the absence of sagittal plane instability or fixed joint deformity [7,60].

Stress fractures of the forefoot commonly present with localized, activity-induced pain and are frequently observed in runners or individuals who have recently increased their activity level [61]. Unlike the plantar-based pain of plantar plate injuries, stress fracture pain is typically dorsal or along the metatarsal shaft [7]. Early radiographs may appear normal, but MRI or bone scintigraphy often reveals cortical disruption and bone marrow edema [7,61]. Point tenderness directly over the metatarsal shaft further aids in distinguishing stress-related injuries from plantar plate pathology [9].

Intermetatarsal bursitis may also mimic the symptoms of neuroma or plantar plate injury, presenting with localized swelling and pain [13,62]. This condition is generally attributed to mechanical friction or overuse and often improves with footwear modification [13,62]. On US, it appears as an anechoic or hypoechoic fluid collection in the intermetatarsal bursa, without associated plantar plate disruption or perineural thickening [62]. Unlike plantar plate tears, bursitis does not result in MTP joint instability or toe deformity [58].

In summary, diagnosing plantar plate injuries is inherently challenging due to significant clinical and radiological overlap with other forefoot disorders, including Morton’s neuroma, MTP joint synovitis, Freiberg disease, stress fractures, and intermetatarsal bursitis. Clinical examination plays a critical role in identifying features such as sagittal plane instability, reduced toe purchase, or joint reducibility—hallmarks typically absent in these mimicking conditions. For instance, a positive drawer or toe purchase test may help differentiate plantar plate insufficiency from isolated neuritic pain or early capsular inflammation. However, these tests may be subtle or inconclusive in early or partial tears. Imaging modalities—particularly MRI and dynamic US—provide essential complementary insights by revealing direct structural disruption, soft tissue integrity, or indirect markers such as capsular thickening and pericapsular edema. MRI is particularly valuable for grading the severity and extent of injury, while US enables real-time evaluation of joint behavior under stress. Ultimately, a combined assessment of clinical and imaging findings is essential to achieve an accurate diagnosis and guide appropriate treatment planning.

The overlapping symptoms, distinguishing features, and characteristic imaging findings of common conditions that may mimic plantar plate injury are summarized in Table 3 to aid in accurate differential diagnosis.

## 6. Summary and Limitations

Plantar plate injuries pose substantial diagnostic challenges due to their subtle clinical manifestations and considerable overlap with other forefoot pathologies. Clinical examination—including the drawer test, toe purchase test, and Kelikian push-up test—remains a critical component of the diagnostic process, though these maneuvers are inherently limited by operator dependency and the absence of standardized criteria. Plain radiography, while unable to directly visualize the plantar plate, can provide indirect signs and is useful in excluding osseous pathologies such as stress fractures or Freiberg disease.

Advanced imaging modalities, particularly MRI and dynamic US, have significantly improved diagnostic precision. MRI offers superior anatomical resolution and high specificity, making it the preferred modality for surgical planning. Dynamic US, on the other hand, serves as a practical and accessible first-line tool, allowing for real-time functional assessment at a lower cost. Emerging techniques, including DECT and dorsiflexion-stress MRI, demonstrate diagnostic promise but currently lack sufficient validation for routine clinical application.

Given the strengths and limitations of each diagnostic tool, clinicians are encouraged to implement a structured, multimodal diagnostic strategy tailored to the individual patient. Continued research and standardization of diagnostic protocols are essential to facilitate earlier detection and more accurate treatment planning, thereby improving clinical outcomes.

This review has several limitations. As a narrative review, it is subject to selection bias and does not incorporate quantitative analyses, such as pooled sensitivity or specificity estimates [63]. Although efforts were made to include high-quality and clinically relevant studies, the absence of a systematic methodology may have resulted in the omission of relevant evidence [64]. Our synthesis may also overrepresent findings from specific institutions or regions, reflecting differences in patient demographics, examiner skill, and imaging equipment that could limit the generalizability of conclusions across diverse healthcare settings [65]. Furthermore, publication bias may have contributed to preferential reporting of positive diagnostic performance, which should be considered when interpreting the summarized evidence [66]. These limitations underscore the importance of cautious application of the findings in varied clinical contexts. Nevertheless, by integrating clinical examination findings with imaging evidence, this review offers a practical framework for improving diagnostic confidence and supporting evidence-based decision-making in the evaluation of plantar plate pathology.

## 7. Clinical Recommendations

Based on the synthesis of current evidence, we recommend a structured diagnostic approach for suspected plantar plate injury that integrates clinical examination with a full spectrum of imaging modalities.

Initial evaluation should include focused history-taking to identify symptom onset, aggravating factors such as barefoot ambulation, high-heeled footwear, or sports activity, and predisposing biomechanical features including elongated second metatarsal, hallux valgus, and pes planus. Targeted physical examination should employ the drawer test, toe purchase test, and Kelikian push-up test to assess sagittal plane stability, dynamic toe purchase, and deformity reducibility, with observation of crossover toe deformity as a late-stage finding.

Patients with positive or equivocal clinical tests should undergo weight-bearing plain radiographs to exclude osseous pathology such as stress fracture or Freiberg disease and to assess indirect signs like MTP subluxation or altered metatarsal parabola. High-frequency dynamic US should be used as the primary advanced imaging tool in most cases, offering accessibility, real-time functional assessment, and high sensitivity, with dynamic maneuvers helping to detect partial tears or functional instability missed on static imaging. MRI is indicated when US findings are inconclusive, for preoperative planning, or when higher specificity and detailed anatomical mapping are required, with dorsiflexion-stress MRI employed if available to improve detection of subtle tears and occult instability. DECT may be considered in rare scenarios where MRI is contraindicated or inconclusive, particularly for collagen mapping, although its role in plantar plate injury remains investigational. In high-risk or complex presentations—such as progressive deformity, recurrent instability, systemic inflammatory disease, or significant biomechanical abnormalities—early integration of imaging into the diagnostic pathway can facilitate timely protective offloading or surgical intervention, helping to prevent progression to irreversible deformity and supporting better long-term functional outcomes.

All imaging findings should be interpreted in conjunction with clinical examination and patient history to avoid misdiagnosis with common mimickers including Morton’s neuroma, MTP synovitis, and intermetatarsal bursitis, ensuring an efficient, accurate, and patient-centered diagnostic process.

## 8. Conclusions

This review consolidates current evidence into a multimodal diagnostic strategy for plantar plate injury, integrating targeted clinical examination with complementary imaging modalities to maximize diagnostic accuracy. Our synthesis goes beyond prior summaries by translating this strategy into a practical, stepwise framework that specifies how and when to apply each modality—ranging from weight-bearing radiographs and dynamic US to MRI, dorsiflexion-stress MRI, and emerging techniques such as DECT—based on clinical presentation and risk profile. By combining the strengths of different modalities, this approach addresses the inherent limitations of any single test, reduces misdiagnosis from overlapping forefoot pathologies, and supports earlier, more targeted intervention.

Future research should focus on refining and validating clinical examination maneuvers, establishing standardized grading systems for both MRI and US, consensus-driven dynamic imaging protocols, and pilot studies on emerging modalities such as DECT and dorsiflexion-stress MRI. These efforts will refine diagnostic pathways, facilitate timely intervention, and improve long-term functional results for patients with plantar plate injury.

## Figures and Tables

**Table 1 diagnostics-15-02188-t001:** Clinical tests for plantar plate injury.

Test	Sensitivity	Specificity	Clinical Relevance	Limitations
Drawer test	80.6%	99.8%	Highly specific indicator of sagittal plane instability; strongly predictive of significant plantar plate tear, especially in high-grade injuries.	Accuracy may decrease in partial tears or in patients with significant guarding due to pain; requires examiner experience for reliable grading; limited utility in very acute presentations with swelling.
Toe purchase test	Not reported	Not reported	Useful for detecting early to moderate functional loss before gross deformity develops; correlates with intraoperative findings.	May yield false positives in conditions such as MTP synovitis or capsulitis; patient cooperation required; lacks widely accepted quantitative threshold for “positive” finding.
Kelikian push-up test	Not reported	Not reported	Assesses deformity reducibility and soft-tissue integrity; helpful for surgical planning	No standardized grading system; relies heavily on examiner subjective judgment.
Crossover toe deformity	8.0%	88.9%	Visual hallmark of advanced plantar plate rupture; indicates high-grade ligament failure.	Represents a late-stage finding, so has very low sensitivity for early injury.

MTP, metatarsophalangeal.

**Table 2 diagnostics-15-02188-t002:** Imaging modalities in plantar plate injury diagnosis.

Test	Sensitivity	Specificity	Clinical Relevance	Limitations
Radiographs	Not reported	Not reported	Useful for detecting indirect signs (MTP subluxation, metatarsal parabola changes) and ruling out osseous pathology (stress fracture, Freiberg disease).	Cannot visualize plantar plate directly; limited in early-stage injury.
Dynamic ultrasound	91–100%	60–95%	High sensitivity; real-time dynamic assessment; inexpensive and widely available.	Operator-dependent; reduced specificity in the presence of synovitis or adjacent soft tissue pathology.
Magnetic resonance imaging (MRI)	89–97%	83–100%	Excellent soft-tissue contrast; detailed anatomical mapping; useful for surgical planning.	Higher cost; longer acquisition time; may miss subtle functional instability without stress positioning.
Dorsiflexion-stress MRI	Not reported	Not reported	Enhances detection of subtle tears and occult instability; improves correlation with intraoperative findings.	Limited availability; patient discomfort in stress position.
Dual-energy CT (DECT)	Not reported	Not reported	Visualizes collagen-rich structures via material decomposition; potential alternative when MRI is contraindicated.	Evidence limited to isolated case reports; not validated in larger, controlled studies.

MTP, metatarsophalangeal; MRI, magnetic resonance imaging; CT, computed tomography.

**Table 3 diagnostics-15-02188-t003:** Differential diagnosis of plantar plate injury.

Condition	Similarities	Differences	Key Imaging Findings
Morton’s neuroma	Forefoot pain; may have Mulder’s click	Neuritic pain localized to webspace; no MTP instability	US: Hypoechoic mass in intermetatarsal space; MRI: T1 low, T2 high signal mass between metatarsal heads
MTP joint synovitis	Swelling, pain at MTP joint	Often involves multiple joints; elevated inflammatory markers	US: Synovial thickening with Doppler hyperemia; MRI: Synovial enhancement
Freiberg disease	Forefoot pain	Typically adolescents/young adults; collapse of metatarsal head	Radiographs: Flattening/sclerosis of metatarsal head; MRI: Subchondral signal changes
Stress fracture	Localized forefoot pain	Worsens with activity; focal tenderness over metatarsal	Radiographs: Often normal early; later cortical break; MRI: Bone marrow edema
Intermetatarsal bursitis	Pain between metatarsal heads	No MTP instability; often improves with footwear modification	US: Fluid-filled bursal sac; MRI: High T2 signal in intermetatarsal space

MTP, metatarsophalangeal; US, ultrasound; MRI, magnetic resonance imaging.

## Data Availability

Not applicable.

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
