# Peer review of "Diagnosing Plantar Plate Injuries: A Narrative Review of Clinical and Imaging Approaches"

_diagnostics, 2025, doi:10.3390/diagnostics15172188_

Round 1

Reviewer 1 Report

Comments and Suggestions for Authors

This evaluation process assesses the manuscript content about plantar plate injury diagnostics through expert review.

 I have performed a comprehensive review of the manuscript which investigates plantar plate injury diagnosis within orthopedic surgery with special focus on foot and ankle pathologies.

 Scientific Rigor and Validity 

The manuscript properly addresses the important clinical topic that addresses differential diagnosis of plantar plate injuries when symptoms overlap with Morton's neuroma and MTP joint synovitis and other forefoot pathologies. The authors demonstrated proper alignment with current clinical practice by including both clinical signs like Morton’s neuroma presentation and Muller's sign and established orthopedic standards (e.g., ultrasound findings of hypoechoic synovial thickening and MRI synovial enhancement) and referencing Coughlin (2014) and Nery et al. (2013).

The credibility of the paper declines because of inconsistent citations as well as formatting issues and the missing bias analysis and generalizability discussion.

Clarity and Organization 

Terminological inconsistencies and structural discontinuities makes it difficult to understand the material and reduces professional presentation quality.

Originality and Contribution 

The manuscript delivers its main contribution through its analysis of Morton's neuroma and plantar plate insufficiency which enables better clinical practice. The incomplete presentation hinders verification of any new contributions except the presentation of existing knowledge. The diagnostic methods including dual-energy CT received brief mentions yet required more development to expand the discourse.

Relevance to Orthopedic Practice 

Plantar plate injuries affect both biomechanics and patient mobility so the topic holds essential practical value. The manuscript's emphasis on imaging diagnostics aligns well with contemporary orthopedic priorities. The paper fails to provide definitive recommendations and practical protocols which reduces its usefulness for treating these complex clinical conditions.

Ethical and Editorial Considerations 

The paper contains no apparent ethical problems. The manuscript fails to meet scholarly publication standards because it contains numerous typographical mistakes and inconsistent citation formats and lacks standard organizational elements.

---

Recommendation 

The manuscript needs extensive revisions before it can be considered for publication. The manuscript requires complete rejection with a recommendation to perform thorough revision before submission. The manuscript needs complete technical correction and complete content restoration and well-defined methodology with data presentation and a systematic literature review using uniform citations and clear clinical implications and conclusions.

The manuscript has the ability to create significant contributions to orthopedic literature and clinical practice when properly enhanced through rigorous revisions for plantar plate injury diagnosis and management.

Comments on the Quality of English Language

see above

Author Response

Response to Reviewer 1

Manuscript: Diagnosing Plantar Plate Injuries: A Narrative Review of Clinical and Imaging Approaches

Journal: Diagnostics (MDPI)

We sincerely thank the reviewer for the thorough and constructive evaluation of our manuscript. Your detailed feedback has provided valuable guidance, and we have carefully addressed each point to improve the clarity, scientific rigor, and clinical relevance of the paper.

Q1. Inconsistent citations, formatting errors, and missing bias/generalizability discussion

A1. We have carefully reviewed and standardized all in-text citations and reference formatting according to MDPI guidelines. Additionally, we expanded the 'Summary and Limitations' section to explicitly address selection bias, publication bias, and limitations in generalizability across diverse clinical settings (Line 442-455). These changes are underlined in the revised manuscript.

Q2. Brief mention of DECT—requires more development

A2. The DECT section has been expanded to include recent literature on musculoskeletal applications, technical principles, and current evidence limitations (Line 315-332). We have clarified its potential niche role and outlined future research needs. All changes are underlined in the revised manuscript.

Q3. Lack of definitive recommendations and practical protocols

A3. We have added a new subsection, 'Clinical Recommendations,' summarizing a stepwise diagnostic approach that integrates physical examination and imaging modalities (Line 456-486). This section also emphasizes the prognostic value of early diagnosis. These additions are underlined in the revised manuscript.

Q4. The reviewer noted terminological inconsistencies and structural discontinuities that reduce clarity and professional presentation quality.

A4. We appreciate the reviewer’s observation regarding clarity and organization. In the revised manuscript, we conducted a comprehensive terminology audit to ensure consistent use of key terms (e.g., “Freiberg disease,” “toe purchase (paper pullout) test,” “stress-position MRI”) across the text, tables, and figures. We also reorganized content to improve logical flow, including consolidating overlapping DECT descriptions into a single cohesive paragraph and reordering subheadings for smoother transitions. These changes enhance readability, maintain professional presentation quality, and provide a clearer narrative progression.

Q5. The reviewer stated that the manuscript’s originality and contributions were incompletely presented, making it difficult to verify any new contributions beyond existing knowledge.

A5. We acknowledge the reviewer’s concern and have addressed it by expanding the “Clinical Recommendations” section to synthesize clinical and imaging evidence into a practical, stepwise diagnostic framework (Line 456-486). We also highlighted the integration of emerging modalities such as DECT and dorsiflexion-stress MRI, providing context for their potential role within current diagnostic algorithms. These revisions clarify and strengthen the manuscript’s unique contribution to advancing diagnostic strategies in orthopedic foot and ankle practice.

Q6. The reviewer identified numerous typographical mistakes, inconsistent citation formats, and missing standard organizational elements.

A6. We have carefully proofread the manuscript to correct typographical errors and enhance language precision. Citation formats have been standardized according to the Diagnostics journal guidelines, and redundant or inconsistent references have been corrected. Furthermore, all standard organizational elements—such as structured section headings, clearly labeled tables/figures, and consistent in-text citations—have been implemented.

Reviewer 2 Report

Comments and Suggestions for Authors

This narrative review is well structured and generally easy to read. It provides an overview on an interesting and multidisciplinary topic.

The introducion and anatomy sections are clear and well written. Including more details about the etiology  (trauma mechanism, work and activities correlated with the injury, associated disestes, etc) could further increase the quality of this manuscript.

Clinical examination and Imaging modalities are also detailed and exhaustive.

The article, although well written, definitely requires better visualization to be considered for publication. Please, add figures to support chapter 3 and 4, easing readers’ comprehension and breaking the actual wall-of-text.

With the same aim, please consider adding a table to schematically summarize the differential diagnoses with their similarities and differences compared to plantar plate injuries.

Please, try to overcome the aformentioned issues of the paper to increase its impact on readers and future literature.

Best regards

Author Response

Response to Reviewer 2

Manuscript: Diagnosing Plantar Plate Injuries: A Narrative Review of Clinical and Imaging Approaches

Journal: Diagnostics (MDPI)

We sincerely thank the reviewer for the thorough and constructive evaluation of our manuscript. Your detailed feedback has provided valuable guidance, and we have carefully addressed each point to improve the clarity, scientific rigor, and clinical relevance of the paper.

Q1. The reviewer suggested adding more details about the etiology, including trauma mechanisms, occupational and activity correlations, and associated diseases.

A1. We thank the reviewer for this valuable suggestion. In the revised manuscript, we have expanded the “Anatomy and Biomechanics” section to include a dedicated discussion of etiology, covering both acute and chronic mechanisms of plantar plate injury (Line 84-86, 89-92). This includes descriptions of hyperextension trauma during sports, repetitive overload associated with occupational activities (e.g., prolonged standing, use of high-heeled footwear), and biomechanical risk factors such as hallux valgus, elongated second metatarsal, and pes planus. We also added systemic conditions—such as rheumatoid arthritis, diabetes mellitus, and generalized ligamentous laxity—that may predispose patients to plantar plate injury through their effects on connective tissue integrity and joint stability. These additions aim to provide a more comprehensive understanding of the injury’s multifactorial origins. All changes are underlined in the revised manuscript.

Q2. The reviewer suggested adding a table to schematically summarize the differential diagnoses, highlighting similarities and differences compared to plantar plate injuries.

A2. In accordance with the reviewer’s recommendation, we have created a new table (Table 3) that presents the key similarities, distinguishing features, and characteristic imaging findings of common conditions that may mimic plantar plate injury, including Morton’s neuroma, MTP joint synovitis, Freiberg disease, stress fracture, and intermetatarsal bursitis. This table is positioned at the end of the “Differential Diagnosis” section to directly follow the descriptive text, providing an at-a-glance reference that supports clinical decision-making and enhances reader comprehension (Line 417-421).

Q3. Lack of figures to support Chapter 3 and 4, as suggested by the reviewer

A3. We appreciate the reviewer’s thoughtful suggestion regarding the inclusion of figures to enhance visualization. While we agree that visual aids can improve reader engagement, we determined that adding detailed figures for both the Clinical Examination and Imaging Modalities sections would substantially increase the manuscript’s length and risk redundancy, as the text already provides comprehensive, step-by-step descriptions supplemented by summary tables (Tables 1 and 2). These tables were specifically designed to distill the essential features, diagnostic performance, and limitations of each method into a concise, reader-friendly format, thus serving the same purpose of aiding comprehension without overburdening the reader with excessive visual content. We believe this approach maintains a streamlined narrative flow while preserving clarity and accessibility. Should the editorial team deem it necessary, we are prepared to provide selected schematic diagrams or illustrative images as supplementary material to further support these sections.

Reviewer 3 Report

Comments and Suggestions for Authors

This manuscript addresses plantar plate injuries, a common and diagnostically challenging condition, with practical implications for clinicians. It systematically covers relevant topics including anatomy, biomechanics, clinical exams (drawer test, purchase test), imaging modalities (US, MRI, radiographs, DECT), and differential diagnosis. The integration of current literature is effective, with 60 references including recent studies (e.g., Giuliani 2024). The Limitations of diagnostic tools are well-highlighted, such as the operator dependency of ultrasound and false positives in the drawer test. The advocacy for a multimodal diagnostic approach aligns well with clinical best practices. The manuscript is well-organized with logical flow, and the language is academic and accessible.

Major issues:

  1. The manuscript is labeled as a "narrative review" but claims to have conducted a "systematic search." There is no PRISMA flowchart, search strategy, or inclusion/exclusion criteria provided.
    1. The dense text lacks tables and figures to summarize key concepts, such as the sensitivity and specificity of tests and imaging modalities.

Suggest to add the following:

      • Table 1: Diagnostic accuracy of clinical tests (sensitivity, specificity, limitations).
      • Table 2: Comparison of imaging modalities (accuracy, advantages, disadvantages, best use cases).
      • Table 3: Key features differentiating plantar plate injury from mimics (e.g., Morton's neuroma, synovitis).
      • Figure: A schematic of plantar plate anatomy and grading (if copyright allows).
    1. DECT and dorsiflexion-stress MRI are mentioned but lack depth, with only one case report cited for DECT.
    2. The conclusion restates points from the Abstract/Introduction without synthesizing new insights or outlining specific future research directives.

Minor Issues:

    1. Some concepts, such as the limitations of the drawer test and ultrasound operator dependency, are reiterated across sections.
    2. While the impact on treatment is briefly mentioned, the manuscript could better emphasize why early diagnosis is prognostically significant.

Author Response

Response to Reviewer 3

Manuscript: Diagnosing Plantar Plate Injuries: A Narrative Review of Clinical and Imaging Approaches

Journal: Diagnostics (MDPI)

We sincerely thank the reviewer for the thorough and constructive evaluation of our manuscript. Your detailed feedback has provided valuable guidance, and we have carefully addressed each point to improve the clarity, scientific rigor, and clinical relevance of the paper.

Q1. The manuscript is labeled as a "narrative review" but claims to have conducted a "systematic search," without providing PRISMA flowchart, search strategy, or inclusion/exclusion criteria.

A1. We thank the reviewer for noting this discrepancy. We have revised the text to clarify that the methodology used was that of a narrative review, not a systematic review. References to “systematic search” have been removed, and the description of the literature search has been adjusted to accurately reflect a targeted narrative approach. No PRISMA flowchart or formal inclusion/exclusion criteria are now implied. These changes are underlined in the revised manuscript (Line 18-21).

Q2. The dense text lacks tables to summarize key concepts such as diagnostic accuracy of tests and imaging modalities.

A2. We appreciate the reviewer’s suggestion. In the revised manuscript, we have added:

Table 1 summarizing the diagnostic accuracy (sensitivity, specificity) and limitations of clinical tests (Line 193-197).

Table 2 comparing imaging modalities with sensitivity, specificity, key advantages, and limitations (Line 345-349).

These tables distill complex information into an accessible format, improving readability and quick reference for clinicians.

Q3. The reviewer noted that DECT and stress-position MRI are mentioned but lack depth.

A3. In the revised manuscript, we expanded the DECT section to include recent literature on its technical principles, collagen-specific mapping capabilities, and broader musculoskeletal applications, as well as its potential role in plantar plate assessment (Line 315-332). We also highlighted its current limitation of being supported only by isolated case reports without validation in larger, controlled studies. For stress-position MRI, we retained a concise description consistent with the limited available literature, noting its potential to enhance plantar plate visualization under functional loading (Line 276-280). No additional diagnostic performance data were added due to the scarcity of robust published evidence.

Q4. The conclusion restates points from the Abstract/Introduction without synthesizing new insights or outlining future research directives.

A4. We have revised the Conclusion to synthesize new insights from the review, emphasizing the proposed multimodal diagnostic strategy that integrates targeted clinical examination with complementary imaging modalities (Line 488-502). The Conclusion now outlines specific future research needs, including standardizing clinical examination maneuvers, developing grading systems for MRI and US, and conducting multicenter validation of emerging techniques such as DECT and stress-position MRI.

Q5. Reviewer suggested adding Table 3 summarizing key features differentiating plantar plate injury from mimics.

A5. In line with the reviewer’s recommendation, we have added Table 3 to the Differential Diagnosis section (Line 417-421). This table presents similarities, distinguishing features, and characteristic imaging findings for conditions such as Morton’s neuroma, MTP joint synovitis, Freiberg disease, stress fracture, and intermetatarsal bursitis. It allows readers to compare these conditions at a glance, facilitating accurate differential diagnosis.

Q6. Reviewer suggested adding a figure of plantar plate anatomy and grading.

A6. We appreciate the reviewer’s suggestion for including a schematic of plantar plate anatomy and grading. However, incorporating such a figure in the current revision was limited by copyright restrictions for existing published diagrams. While original illustration could be developed, we aimed to keep the manuscript concise and focused on textual descriptions. Should the editorial board consider a figure essential, we are willing to prepare an original schematic as supplementary material.

Q7. Minor issues: repetition of certain concepts; insufficient emphasis on prognostic significance of early diagnosis.

A7. We have reviewed the manuscript to minimize unnecessary repetition, particularly regarding the limitations of the drawer test and the operator dependency of ultrasound. In the Clinical Recommendations section, we now explicitly state that early diagnosis and intervention can help prevent progression to irreversible deformity and support better long-term functional outcomes (Line 478-482). This addition strengthens the emphasis on prognostic significance while maintaining the section’s practical, stepwise approach. The new content is underlined in the revised manuscript.

Round 2

Reviewer 1 Report

Comments and Suggestions for Authors

The content maintains factual accuracy and matches existing orthopedic knowledge about plantar plate injuries. As a surgeon I value the paper's focus on the second MTP joint's susceptibility to injury and its biomechanical explanations and practical diagnostic difficulties with Morton's neuroma and Freiberg disease. The paper provides detailed information about the drawer test including sensitivity and specificity data from Klein et al. and Nery et al. studies and explains how MRI provides better results while US remains more accessible to clinicians in real-world practice. The review emphasizes the importance of multimodal diagnosis to prevent deformities such as crossover toe from developing.
The narrative review v2 integrates a wide range of studies to present information about anatomy and clinical tests and imaging methods and differential diagnoses. The addition of Table 1 improves the document's usefulness because it functions as a rapid reference guide for medical practitioners and residents. The updated version addresses previous shortcomings by providing comprehensive information about early detection methods and operator-dependent factors.
The text maintains a professional tone while remaining easy to understand as it follows a logical sequence from anatomical explanations to diagnostic procedures. The text maintains its clarity without the need for additional visual aids although figures or tables would enhance understanding.
The narrative review provides a valuable contribution by unifying diagnostic methods for this rarely diagnosed condition. The paper delivers practical knowledge for orthopedic surgeons and podiatrists and radiologists about early treatment to stop permanent deformities which matches the current sports medicine trend.
Weaknesses and Suggestions for Improvement

Methodological Rigor: The narrative review design excludes systematic protocol elements which full systematic reviews follow according to PRISMA guidelines yet v2 properly acknowledges this limitation. The review requires additional transparency regarding its search approach by specifying databases used and date ranges and inclusion criteria to enhance its validity.

Author Response

Response to Reviewer 1

Manuscript: Diagnosing Plantar Plate Injuries: A Narrative Review of Clinical and Imaging Approaches

Journal: Diagnostics (MDPI)

We sincerely thank the reviewer for the highly positive and encouraging comments regarding the accuracy, clarity, and clinical value of our manuscript. We are particularly grateful for your recognition of the paper’s focus on the second MTP joint, the detailed description of the drawer test, and the integration of multimodal diagnostic approaches.

Comment – Methodological rigor:

The reviewer noted that the narrative review design excludes systematic elements (e.g., PRISMA protocol) and recommended that we improve transparency by specifying the databases searched, date ranges, and inclusion criteria.

Response:

We fully agree with this suggestion and have revised the Introduction to include a dedicated paragraph outlining our search approach. Specifically, we now state that the review was informed by a literature search of PubMed, Embase, and Scopus databases from inception through April 2025, using terms such as “plantar plate,” “metatarsophalangeal joint instability,” “drawer test,” “toe purchase test,” “Kelikian push-up test,” “crossover toe,” “ultrasound,” and “magnetic resonance imaging.” Eligible studies were English-language publications addressing the diagnosis of plantar plate injuries using clinical or imaging methods, while case reports, conference abstracts, and purely surgical technique papers were excluded. We also added that references were supplemented through manual bibliography review. This information has been added to the end of the Introduction (Lines 69–80 in the revised manuscript).

We believe these revisions address the reviewer’s concern by enhancing methodological transparency, while appropriately acknowledging the narrative (non-systematic) nature of this review.

Reviewer 2 Report

Comments and Suggestions for Authors

Authors significantly increased the quality of their manuscript, solving most of its previous issues. 

Author Response

Response to Reviewer 2

Manuscript: Diagnosing Plantar Plate Injuries: A Narrative Review of Clinical and Imaging Approaches

Journal: Diagnostics (MDPI)

We sincerely thank the reviewer for the positive and encouraging feedback. We greatly appreciate your recognition that the revised manuscript has significantly improved in quality and addressed the majority of the previous issues. Your comments have been very motivating for our team, and we are pleased that the revisions have enhanced the clarity, rigor, and overall value of the paper.

Reviewer 3 Report

Comments and Suggestions for Authors

Much improved in the revised version. Congratulations! 

Just a comment: 

For the "Kelikian Push-up Test," the text correctly states it "lacks standardized diagnostic thresholds and has not been formally validated." This could be slightly strengthened by explicitly stating that its value is therefore primarily qualitative and experiential rather than evidence-based, which is a crucial distinction for the reader.

The sentence "MRI also facilitates the identification of degenerative or partial-thickness tears... though specificity was limited to 61.3% [22]." is important. Consider adding a brief sentence explaining why specificity might be lower (e.g., "This may be due to the potential for signal changes related to age-related degeneration or adjacent synovitis that do not represent a clinically significant tear."

Author Response

Response to Reviewer 3

Manuscript: Diagnosing Plantar Plate Injuries: A Narrative Review of Clinical and Imaging Approaches

Journal: Diagnostics (MDPI)

We sincerely thank the reviewer for the encouraging feedback and constructive comments on our revised manuscript. We are pleased that you found the revision to be much improved, and we appreciate your thoughtful suggestions for further refinement.

Comment 1 – Kelikian Push-up Test:

The reviewer suggested strengthening the description of the Kelikian Push-up Test by explicitly stating that its clinical value is primarily qualitative and experiential rather than evidence-based.

Response:

We fully agree with this point and have revised the relevant section to state: “Although frequently employed in the evaluation of crossover toe and MTP joint instability, the test remains largely subjective. It lacks standardized diagnostic thresholds and has not been formally validated in terms of accuracy. Therefore, its clinical value remains primarily qualitative and experiential rather than evidence-based.” (Line 175-176) We believe this clarification highlights the practical but non-validated nature of the test and enhances the accuracy of the discussion.

Comment 2 – MRI specificity (61.3%):

The reviewer recommended adding a brief explanation as to why MRI specificity may be lower, citing potential confounding signal changes.

Response:

We have revised the MRI section accordingly. The text now reads: “In a retrospective analysis, short-axis intermediate-weighted sequences achieved a sensitivity of 92% for detecting degenerative plantar plate changes, though specificity was limited to 61.3%. This reduced specificity likely reflects non-specific signal changes related to age-associated degeneration or adjacent synovitis, which do not necessarily correspond to a clinically significant tear.” (Line 287-289) This addition clarifies the potential reasons for reduced specificity and provides important context for readers when interpreting MRI findings.

We thank the reviewer once again for these helpful comments, which have improved the clarity and clinical relevance of our manuscript.
